# Perspectives and Implications of Coanda Effect in Aneurysms

**DOI:** 10.3390/brainsci13060966

**Published:** 2023-06-19

**Authors:** Vicentiu-Mircea Saceleanu, Razvan-Adrian Covache-Busuioc, Luca-Andrei Glavan, Antonio-Daniel Corlatescu, Alexandru Vlad Ciurea

**Affiliations:** 1Neurosurgery Department, Sibiu County Emergency Hospital, 550245 Sibiu, Romania; vicentiu.saceleanu@gmail.com; 2Department of Neurosurgery, “Lucian Blaga” University of Medicine, 550024 Sibiu, Romania; 3Neurosurgery Department, “Carol Davila” University of Medicine and Pharmacy, 020021 Bucharest, Romania; glavan.luca@gmail.com (L.-A.G.); antonio.corlatescu@gmail.com (A.-D.C.); prof.avciurea@gmail.com (A.V.C.); 4Neurosurgery Department, Sanador Clinical Hospital, 70000 Bucharest, Romania; 5Romanian Academy, Calea Victoriei 125, 010071 Bucharest, Romania

**Keywords:** Coanda effect, aneurysm, hemodynamics, Willis Polygon, pathology

## Abstract

It is yet unknown how the formation of an aneurysm inside the human body occurs. Thus, understanding and analyzing the Coanda effect will result in a better overview of the overall fluid mechanics that develop inside such a structure, leading not only to better treatment plans, but also to diminished postoperative risks. This paper presents how the fluid behaves in this situation, and takes into consideration how this physical phenomenon influences the hemodynamics inside numerous anatomical regions, located in the central nervous system, where aneurysms usually develop. Analyzing the three main areas in which cerebral aneurysms form, the Coanda effect can potentially lead to the rupture of the aneurysm by changing the blood flow trajectory; this should be taken into consideration when choosing a treatment plan, especially in postoperative care. In addition, there are other factors that can influence the evolution of an aneurysm, such as its shape, size, localization and the patient’s health condition. Understanding and analyzing the Coanda effect will result in a better overview of the overall fluid mechanics that develop inside such a structure, leading not only to better treatment plans, but also to diminished postoperative risks.

## 1. General Data

### 1.1. Historical Facts

Henri Coanda was born in Bucharest on the 7th of June 1886. His early education consisted mostly of military training due to the fact that his father was a general in the Romanian Army. At just 19 years of age, he constructed his first prototype of a rocket-powered airplane. Later that decade, Henri Coanda created the “Coanda 1910”, an airplane without a propeller, following his graduation from “ Institut Supérieur de l’Aéronautique et de l’Espace”.

Through the years, he invented other aircrafts, including “Coanda 1911” and “Bristol-Coanda”, both with propellers, and “Coanda 1916”, which he invented while working at “Saint-Chamond” in Saint Denis.

Coanda’s inventions not only had a great impact on further discoveries in the aeronautical domain, which were used in the First World War, but also in fields such as medicine, architecture, transport and construction, among others.

In 1930 [1], he obtained a French patent for the observation and description of the “Coanda Effect”, a physical phenomenon that, besides medicine, has numerous appliances in fields such as aeronautics, fluids, and meteorology. Henri Coanda died on the 25th of November 1972.

### 1.2. The Physics of the Coanda Effect

The Coanda effect is a physical phenomenon that describes the tendency of a fluid or gas to attach to a surface after passing through an asymmetrical narrowing, exerting force on neighboring fluid or gas. We believe that the Coanda effect is overlooked when taking into consideration the formation and development of numerous types of aneurysms, and also the postsurgical events that can occur after clipping an aneurysm. This physical phenomenon is of great importance, both when the blood passes through a narrowing of the lumen, as well as when the blood travels through an arterial passage that is physiological or pathologically enlarged in comparison to the rest of the artery.

Recognizing the effect of the Coanda effect in aneurysms is vitally important for several reasons. First, it provides insights into its genesis. When blood flow is attached to irregular surfaces within vessel walls due to this effect, the shear stress levels may be altered, leading to remodeling that contributes to aneurysm formation. Furthermore, understanding these effects can assist with identifying individuals at a greater risk of aneurysm development and suggest preventative measures that should be implemented accordingly.

From a medical standpoint, a narrowing inside the body develops, for example, when a cholesterol plaque is created. In addition, being a protuberance (convex structure), it will divert the blood flow on the opposite side of the artery, creating a pressure gradient between the blood jet and the rest of the lumen. When this phenomenon develops, the blood flow will become adherent to the wall, creating further pressure on the afferent wall while also recruiting adjacent liquid particles into the stream. It is important to mention that the Coanda effect occurs for Reynold’s numbers as small as 50, meaning that it is present both in laminar and turbulent flows. Also, the Coanda effect is viable for both high and extremely low pressure [2]. It is important to note that if the narrowing of the vessel is developed close to a junction of an artery, the blood flow will tend to become asymmetrical, thus creating a flux gradient between the two ramifications. The effect on the boundary wall makes the blood tend to stick to the wall and create a pressure gradient, with the jet adhering more to a ramification; this results in the possibility of some blood joining the ramification where the pressure is higher, creating a retrograde flow (Figure 1).

According to [2], the protuberance will create an asymmetry inside the blood vessel, resulting in the jet being diverted to the opposite wall; thus, the jet will create, according to the Coanda effect, a pressure gradient that will cause a retrograde flow from the A2 artery to the A1 artery, a phenomenon that can lead to dangerous ischemic damage.

However, when the surface of the blood vessel is concave the phenomenon will become opposite to the one previously described. The blood jet will tend to attach to the concave surface due to the low pressure inside the pit, thus creating small-scale swirls that exert pressure on the arterial wall. This event will divert the blood flow on the wall with the concavity, and, according to the Coanda effect, will promote neighboring liquid to join the newly created jet. This will, once again, result in an asymmetry of the blood flow if this event happens before a junction (Figure 2).

## 2. Typical Aneurysms

An aneurysm is a vascular structure that implies a weakening of a blood vessel (with all its layers) that can be localized along the route of both arteries and veins. It takes form as a dilatation of the blood vessel and can be classified based on various factors such as location (abdominal, thoracic, cerebral or peripheral), shape (fusiform, saccular), size (small, large, giant) and cause (congenital, infectious, gained, tumorous, dissecting, traumatic). A special classification of aneurysms is based on their association with the surrounding vessels; this has been proven to be useful in treating cerebral aneurysms [3].

## 3. Typical Form of Cranial Aneurysms

Regarding cranial aneurysms, based on their form, they can be either fusiform or saccular.

Fusiform—This type of intracranial aneurysm is represented by cylindrical dilatations formed at a relatively short distance on the vessel and is usually associated with atherosclerosis and a high blood pressure [4].

Coanda effect in fusiform aneurysm—In this case, the Coanda effect appears on the blood vessel wall due to the presence of concavities. This physical phenomenon will promote the stream to behave in a certain way: a part of the jet will be bound to one wall, another part will be bound to the other side, and a part of the jet will flow through the center of the blood vessel (in the case of a bilateral fusiform aneurysm). The volume of the stream that will be bound to one side or another can be influenced by a number of factors, such variations in the pressure, a more specific level of contraction force in the walls, the number of particles involved, the temperature of the environment, the flow rate, the viscosity of the blood, and also the size and shape of the aneurysm [5]. Inside these types of concavities, micro swirls will be created, applying continuous pressure upon the vessel’s walls. In time, this will lead to an overall increase in the size of the aneurysm.

Another important factor is how the aneurysm is clipped intraoperatively; if the aneurysm is clipped too tight or the placement is not perfect, the Coanda effect will produce a deviation in the jet and an asymmetrical blood flow. This effect causes the blood to follow the curved walls of the vessel, contributing to an increase in the pressure inside the blood vessel [2]; as a result, it will progressively expand, potentially leading to a potential rupture. The vascular complications are caused not only by an increase in the pressure, but also by the fact that the Coanda effect causes the blood to stagnate in certain parts of the aneurysm, resulting in the formation of thrombi. In addition to the progression of the aneurysm, taking the Coanda effect into consideration can change the treatment approach, especially when using endovascular techniques. In this case, the catheter is guided by this physical phenomenon on the wall and can be used to direct the blood flow away from the aneurysm. In this way, the risk of rupture is lowered and the normal jet stream is restored [6].

Saccular—This type of intracranial aneurysm is represented by small sac-like dilatations in an artery that are usually associated with the weakening of the blood vessel wall caused by an increase in the hemodynamic pressure, as well as the distension of the artery [7]. It can also be associated with tumorous structures or with infections such as tuberculosis [8].

### Coanda Effect in Saccular Aneurysms

Our understanding of saccular aneurysms develops from convex structures inside the blood vessels. As mentioned before, convex structures appear inside the body when, for example, a cholesterol plaque forms inside an artery. The blood will interact with the convex structure and stick to the opposite wall of the artery, according to the Coanda effect. Subsequently, the resistance of the afferent wall decreases and, in time, will lead to the formation of a concavity. As previously stated, due to the Coanda effect, a concavity will cause the blood flow to adhere to it, a vicious cycle that will, in time, lead to the formation of a saccular aneurysm.

It Is mandatory to perform surgery on an aneurysm under conditions of hypotension; thus, after clipping an aneurysm, if the clip is placed too tightly and too close to the base of the aneurysm, post-operatively we may identify a protrusion inside the vessel, generating other life-threatening complications.

## 4. Special Types of Brain Aneurysm and How They Are Affected by the Coanda Effect

Serpentine (Giant)—This type of aneurysm is a giant aneurysm that is formed by an evolving saccular or fusiform type. It has a complex form and a sinuous vascular channel. From an angiographic perspective, the features presented by a serpentine aneurysm are as follows: a diameter greater than 25 mm; an undulating intra-aneurysmal vascular channel; and a common association with partial thrombosis [9]. The Coanda effect appears when the blood jet flows through the aneurysm and can cause the stream to bind to the walls of the blood vessel. As a result, areas of stagnating blood will appear, thus increasing the risk of blood clots forming within the aneurysm. The micro swirls that form within the environment of the blood vessel will also contribute to increasing the pressure of the jet stream and to its deviation on the opposite wall from the one on which the aneurysm is situated. There are a number of techniques that address the Coanda effect in serpentine aneurysms, such as surgical clipping or using stents to redirect the blood flow and minimize the risk of blood clots appearing; these methods are very relevant considering the increased risk of stroke compared to other types of aneurysms.

## 5. Coanda Effect Inside the Main Aneurysmal Areas of the Willis Polygon

Anterior communicating artery

Aneurysms appear commonly at the junction between the anterior communicating artery, the pericallosal artery and the anterior cerebral artery. Recent studies [10] have shown that for these types of aneurysms, the most common approach is clipping, because endovascular treatment is often too inefficient when performed in this segment. The parallel clipping of these types of aneurysms will lead to the formation of a protuberance in the walls of the anterior cerebral artery; this protuberance will divert the majority of the flow either through the pericallosal artery, through the anterior communicating artery, or into the opposite pericallosal artery. Therefore, the Coanda effect can lead to serious ischemic damage in postoperative care (Figure 3).

2.Bifurcation of Internal Carotid Artery in Anterior Cerebral Artery and Middle Cerebral Artery

ICA aneurysm is a type of cerebral aneurysm that develops in the internal carotid artery; this is one of the main arteries that supplies blood to the brain and is mostly saccular, considering the shape. The junction in this region is formed by the internal carotid artery, anterior cerebral artery and the middle cerebral artery, thus forming a T-shape with an alpha angle of approximately 180 degrees. The constriction of the wall appears on the internal carotid artery, but it has to be at a very high level in order for the Coanda effect to appear; however, if it appears, the blood flow will adhere to the opposite wall of the artery and will be lead to the anterior cerebral artery [2]. The main risk is represented by the progression of the aneurysm, which can lead to rupture as well as the risk of blood clots forming inside the aneurysm; therefore, the localization of the aneurysm at this bifurcation is relevant and can influence the form treatment. Lower aneurysms can be clipped using any technique, while higher ones need to be clipped at the right angle; otherwise, the Coanda effect can appear, leading to other treatment plans needing to be addressed in order to minimize the risk that arises with the occurrence of this phenomenon, as well as to a deviation in the blood flow. Other factors that can influence the manifestation of fluid mechanics disturbances, as well as the Coanda effect, are the size, location, and shape of the aneurysm, as well as the patient’s age, overall health state, and other medical conditions (Figure 3).

3.Bifurcation of Basilar artery in the two Posterior Cerebral arteries

A basilar artery aneurysm is a type of cerebral aneurysm that appears in the basilar artery, which is a large blood vessel inside the brain that supplies the brainstem and the cerebellum. These aneurysms are extremely rare and are mostly saccular. The junction in this region is Y-shaped, which makes it a textbook case of the Coanda effect, which causes the blood flow to deviate to the opposite wall of the aneurysm into the posterior cerebral artery. In this case, the Coanda effect plays a most significant role in determining the blood stream’s trajectory change, as well as the progression of the aneurysm. Treatment options include clipping, flow diversion and endovascular coiling, reducing the main risk of rupture in this area. In some cases, for example, after clipping, a convection can appear below the clipping area, which can determine the occurrence of another aneurysm; in this instance, multiple treatment plans should be applied. The treatment plan also can depend on other factors, such as the shape or size of the aneurysm, the patient’s health condition and the risk of rupture (Figure 3).

## 6. Discussion

It is of great importance, when looking at the human body as a whole, to take into consideration all the physical phenomena that occur naturally inside it. The tendency of the fluid to attach to a part of the artery can lead to a deviation in the blood flow, resulting in serious ischemic damage in the absence of thrombosis [11]. This effect can explain both how aneurysms are created, and also how they develop. Using actual methods, such as Doppler echocardiography, computing tomography and magnetic resonance imaging, we can detect the Coanda effect and use different treatment plans in order to minimize the risks that can occur.

Understanding how the Coanda effect influences hemodynamics might be an invaluable asset when performing surgical aneurysm treatments and postoperative interventions. First and foremost, understanding this phenomenon allows medical practitioners to approach aneurysm clipping procedures with greater caution and precision. Taking into account the intricate dynamics of blood flow during such procedures, surgeons can make more informed decisions during clipping processes, thus ensuring adequate aneurysm treatment while mitigating the risks associated with complications.

Further study of the Coanda effect in aneurysms is essential to our comprehension of their complex hemodynamics. Although its influence on blood flow within aneurysms has long been recognized, more needs to be explored and understood regarding this phenomenon’s role in aneurysm formation, progression, and rupture.

Incorporating knowledge of the Coanda effect into clinical practice guidelines will allow healthcare providers to benefit from consistent protocols and recommendations for managing aneurysms. Such guidelines may outline considerations to account for its impact, enabling physicians to tailor treatment strategies according to this knowledge, thus furthering consistency, uniformity, and improving treatment outcomes overall.

Overall, an extensive examination of the Coanda effect in aneurysms could significantly advance our knowledge. By deciphering how it affects aneurysm hemodynamics and discovering its underlying mechanisms, a detailed investigation could pave the way for improved diagnostic, therapeutic and preventive strategies that will ultimately benefit those suffering from these complex vascular conditions.

## Figures and Tables

**Figure 1 brainsci-13-00966-f001:**
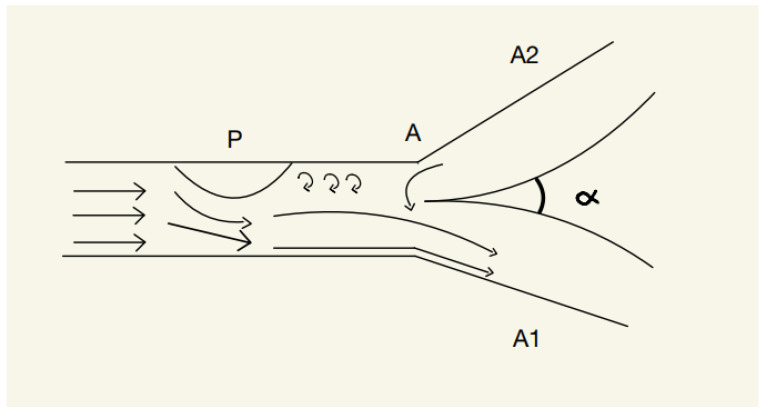
P—Protuberance, A—point of junction, A1—artery 1, A2—artery 2.

**Figure 2 brainsci-13-00966-f002:**
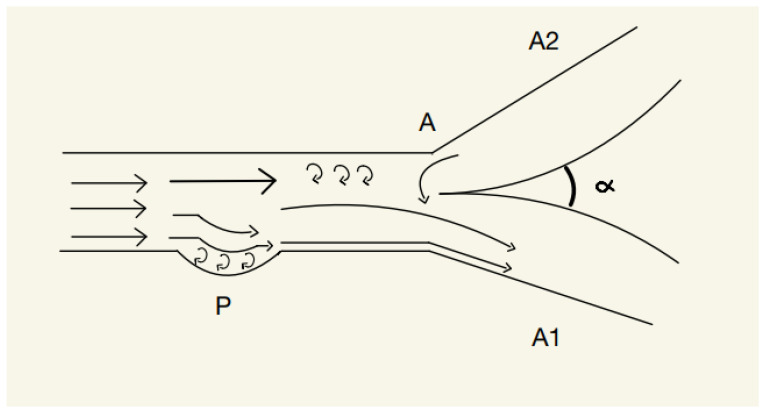
P—Concavity, A—point of junction, A1—artery 1, A2—artery 2.

**Figure 3 brainsci-13-00966-f003:**
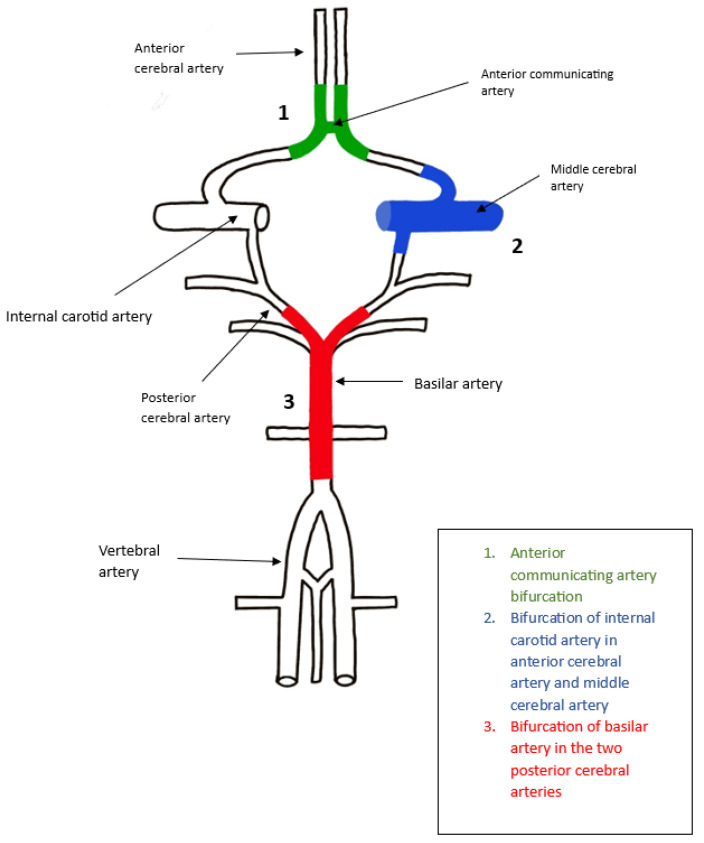
Coanda Effect Inside the Main Aneurysmal Areas of the Willis Polygon.

## Data Availability

All Data is available on PubMed.

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
