# Peer review of "Perspectives and Implications of Coanda Effect in Aneurysms"

_brainsci, 2023, doi:10.3390/brainsci13060966_

Round 1
Reviewer 1 Report
This opinion article is well written and exposes clearly the coanda effect in aneurysms potentially located in the three main areas of the central nervous system. However, I do not find the novelty in the present manuscript since the authors published a previous work in 2019 about a similar topic.
From my point of view the information contained in the article regarding the recommendations for surgery and postoperative treatment should be included as part of a clinical practice guidelines publication to handle brain aneurysms.
No comments
Author Response
Dear Reviewer,
First of all, Thank You for all the guidance and help provided!
The short manuscript (2 pages and half) published in 2019 was structured as a very basic hypothesis, based(the manuscript) on a conference paper from a congress earlier that year.
We strongly assure you that there are no duplicates.
The new perspectives are represented by a fresh approach adopted to examine the fluid dynamics (Coanda effect) aspects of aneurysms, along with a focus on their most common types in terms of shape and location.
Regarding your second suggestion "From my point of view the information contained in the article regarding the recommendations for surgery and postoperative treatment should be included as part of a clinical practice guidelines publication to handle brain aneurysms." - We will strongly take that into consideration! - As we emphasised in our revised version.
Best regards!
Reviewer 2 Report
The paper presented is a sort of narrative review focused on the medical impact of the Coanda effect on the formation, progression and medical implication of cerebral aneurysms. The paper is well-written and centred on a particular hemodynamic aspect that has an important medical aspect in terms also of procedural and treatment implications.
I don't really understand the type of article that is presented. It is neither an original article nor a review. The discussion section is so limited and has to be implemented. I would ask the authors to revise the manuscript with its re-organization in order to become a real narrative review.
Moderate English revision is mandatory
Author Response
Dear Reviewer,
First of all, Thank You for all the guidance and help provided!
We are glad that you appreciated our work!
Regarding: "I don't really understand the type of article that is presented. It is neither an original article nor a review. The discussion section is so limited and has to be implemented. I would ask the authors to revise the manuscript with its re-organization in order to become a real narrative review. " - The manuscript is not a review, nor an original article - it is an "Opinion" type of article, which consists of a 2500 words minimum short and accurate description of a phenomenon(in our case).
Regarding the Discussion section, it has been improved and we hope it would be satisfactory.
Best regards!
Reviewer 3 Report
I read with great attention and interest the paper entitled "Perspectives and implications of Coanda Effect in aneurysms"
Although the subject treated by the Authors may be of some interest, I must admit that the paper (at least in its current form) does not present any new elements.
In particular, in my opinion the scientific/experimental data on which the (albeit suggestive) Authors' hypotheses are based
Author Response
Dear Reviewer,
First of all, Thank You for all the guidance and help provided!
The manuscript is not intended to be a review, nor an original article, but an "Opinion" type of article, as it is registered - regarding the "In particular, in my opinion the scientific/experimental data on which the (albeit suggestive) Authors' hypotheses are based" suggestion.
Secondly, "Although the subject treated by the Authors may be of some interest, I must admit that the paper (at least in its current form) does not present any new elements." - The topic approached (Coanda effect in aneurysms) is not widely explored, (there are less than 10 papers published in the last 60 years regarding it), hence the lack of actual hard statistical data.
As the type of article we have chosen (Opinion), it is an "Opinion", but we will improve its weak spots such as the Discussion section and its Introduction.
Best regards!
Round 2
Reviewer 1 Report
I have no further comments
Reviewer 2 Report
After Authors' revision, the paper has increased its quality.
Reviewer 3 Report
I personally feel that the paper has not really been improved through the review process.
Although I believe that an opinion is an opinion, I would like to emphasize that an opinion, to be published, must be new and must enrich the scientific community.